# OpenReview forum: "ReBiSA: Data Reweighting with Bilevel Optimization for Safety Alignment"
_ICLR.cc/2026/Conference — ICLR 2026 Conference Withdrawn Submission_

### Official Review · Reviewer_Fhgn · 2025-10-26

**Soundness:** 3
**Presentation:** 2
**Contribution:** 2
**Rating:** 4
**Confidence:** 5

**Summary:**

This paper proposes ReBiSA, a bilevel data-reweighting method for safety alignment that learns a tiny 2-layer MLP to map each sample’s training loss to a weight in [0,1]. The inner problem minimizes the weighted training loss, while the outer problem minimizes a safety validation objective; to avoid expensive implicit differentiation, the authors adopt a penalty-based single-level reformulation with first-order updates. The pitch is that learning a function over loss values (rather than per-example weights) is more scalable and transferable across datasets/models, yielding good safety gains with modest compute.

**Strengths:**

ReBiSA is a clean extension of bilevel reweighting to the safety setting (SEAL). The two-layer MLP over per-sample loss is simple yet expressive, and the penalty-based relaxation removes the need for Hessians/implicit-diff, keeping updates first-order and practical for LLMs. The paper argues this functional reweighting is dataset-size independent and can be transferred to other datasets/backbones with negligible overhead beyond computing per-sample losses, which are already needed for training. Empirically, the method compares favorably to fixed reweighting rules and per-sample parameterizations while reporting lower optimization overhead.

**Weaknesses:**

1. The MLP still operates per sample (on each sample’s loss), so the runtime/IO pattern scales with batch size and requires per-sample bookkeeping; the claim of negligible overhead would benefit from detailed wall-clock and throughput reporting under realistic LLM finetunes.
2. Why not skip learning a reweighter entirely and do per-sample filtering with a guardrail model (or a safety scorer) at training time? The paper should compare to a strong guardrail-filtered SFT baseline where unsafe samples are down-weighted/removed using an off-the-shelf safety classifier.
3. Relatedly, Shape it Up! Restoring LLM Safety during Finetuning via STAR-DSS provides token-level safety scores without training an auxiliary MLP; please position ReBiSA against STAR-DSS both methodologically and empirically
4. Table 1 is missing key baselines: Antidote: Post-fine-tuning Safety Alignment for Large Language Models Against Harmful Fine-tuning Attack, Lisa, Constrained SFT (Safety Alignment Should Be Made More Than Just a Few Tokens Deep), SafeLoRA, and Vaccine. These are widely cited in the safety-patching/post-SFT literature and necessary for a fair comparison.
5. The paper contrasts itself with per-sample weight bilevel methods (e.g., SEAL: Safety-Enhanced Aligned LLM Fine-tuning via Bilevel Data Selection) mostly on parameter count; it should also report accuracy/safety trade-offs and transfer vs SEAL under identical data and budgets to make the scalability claim compelling.
6. The method uses loss as a proxy for (un)safety. That proxy can be unreliable under distribution shift (e.g., benign but hard items with high loss may be down-weighted, and unsafe but easy items may slip through). Please analyze failure modes of loss-only signals or augment with explicit safety features.
7. Per-sample reweighting does not address semantic mixing within a single example (samples containing both safe and unsafe spans). Attacks like No, of Course I Can! Deeper Fine-Tuning Attacks That Bypass Token-Level Safety Mechanisms and the rejection-sampling weaknesses discussed in Shape it Up! show that span-/token-level safety control is needed; a scalar weight per sample cannot capture intra-sample semantics.
8. The paper claims transferability of the learned MLP across datasets/backbones; this is asserted but lightly evidenced. Please add ablations that (i) train the MLP on one safety domain and test on another, and (ii) freeze the MLP on one backbone and apply to a different family, reporting any degradation.
9. The bilevel objective is approximated by a penalty; please provide sensitivity to the penalty coefficient and confirm stability across seeds, since the trade-off between validation safety loss and weighted training loss is central.
10. Threat-model coverage is limited: there’s no adaptive attacker who meta-poisons data to spoof the loss -> weight mapping (e.g., making unsafe samples appear low-loss) or inflates harmful rank so simple down-weighting is insufficient.

**Questions:**

See above

---

### Official Review · Reviewer_egC1 · 2025-11-01

**Soundness:** 3
**Presentation:** 3
**Contribution:** 2
**Rating:** 4
**Confidence:** 3

**Summary:**

This paper introduces ReBiSA, a bilevel optimization-based data reweighting framework designed to enhance the safety alignment of large language models (LLMs) during fine-tuning. The core of the method is a lightweight MLP-based reweighting network that dynamically assigns a scalar weight to each training sample based on its loss. These weights are learned by optimizing an outer-level objective on a validation set consisting entirely of safe instances, which guides the network to prioritize safe data and suppress unsafe examples in the training set. Compared with prior bilevel methods that assign a unique parameter to each sample, ReBiSA has a fixed number of parameters, making it scalable. Once trained on a small dataset or a small model, the network can be applied to different datasets and models. Results demonstrate that ReBiSA consistently improves safety metrics across multiple test sets. Further analysis shows it effectively reduces the proportion of unsafe data among the highest-weighted samples.

**Strengths:**

1. The idea of using a small MLP to map loss to safety-aware weights is novel and highly practical. ReBiSA avoids assigning independent parameters to each sample or relying on large external models, significantly improving computational efficiency and scalability.
2. The experimental results show that ReBiSA can be transferred across dataset scale and model scale. The ability to train once on a small setup and deploy on larger datasets and without retraining is a powerful feature for real-world applications, which may improve the method's cost-effectiveness and scalability compared to per-sample parameter methods.

**Weaknesses:**

1. **The evaluation lacks an assessment of the model's general capabilities.** The paper demonstrates improvements in safety but does not investigate the potential impact of ReBiSA on the model's general instruction-following ability or helpfulness. This is critical because the reweighting process is applied to the entire training set, not just the unsafe examples. Consequently, the learning dynamics for benign data are also altered. It is posible that down-weighting high-loss samples, which may include complex but safe instructions, could degrade the model's performance on challenging, benign tasks. It is suggested to evaluate ReBiSA on utility benchmarks to determine whether the safety improvement comes at the cost of reduced general utility, which is a core consideration in real-world deployment.

2. **The claims of transferability are not fully explored across diverse data domains and model architectures.** The experiments show effective transfer across data scale and from a smaller model size within the same Llama family. However, the scalability and robustness also require testing in more challenging transfer scenarios. For instance, can a reweighting network transfer across different datasets and  model architectures (e.g., from a Llama-based ReBiSA to a Mistral). Establishing cross-domain and cross-architecture transferability would significantly strengthen the claim that ReBiSA learns a fundamental, model-agnostic principle of safety-aware weighting.

3. **The analysis lacks a thorough investigation of key hyperparameters and data composition.** The performance of ReBiSA is related to factors that are not deeply analyzed. Firstly, the penalty coefficient $\gamma$ controls the crucial balance between the safety signal from the validation set and the reweighted training loss. A sensitivity analysis of $\gamma$ would reveal how this trade-off influences both safety and general utility. Secondly, the method's robustness to the ratio of safe-to-unsafe data in the training dataset is unexplored. It is unclear if ReBiSA remains effective when this ratio is higher, which is vital for understanding its applicability to datasets with different levels of pre-filtering.

**Questions:**

It is well-known that the safety of LLMs can decline even when trained on a fully benign dataset [1]. If the dataset is entirely benign, how does ReBiSA's safety performance compare? Specifically, how does it compare to the base model (i.e., before training) and Standard SFT?

[1] Fine-tuning Aligned Language Models Compromises Safety, Even When Users Do Not Intend To! ICLR 2024.

---

### Official Review · Reviewer_bpMd · 2025-11-01

**Soundness:** 2
**Presentation:** 2
**Contribution:** 2
**Rating:** 2
**Confidence:** 4

**Summary:**

This paper proposes ReBiSA: a data reweighting framework to improve the safety alignment of LLMs while avoiding the high costs associated with traditional methods like reinforcement learning or large auxiliary models. The core of the method is a lightweight Multi-Layer Perceptron (MLP) that learns to map the training loss of each data sample to an importance weight. This is achieved via a bilevel optimization framework: an inner loop fine-tunes the LLM on a training set of mixed-quality data using the weights provided by the MLP, while an outer loop updates the MLP's parameters to minimize the LLM's loss on a curated, safe-only validation set. This process effectively *teaches the MLP to automatically down-weight unsafe data* and *up-weight safe data* during fine-tuning. The authors demonstrate that this approach not only improves safety but is also highly efficient and transferable, as a reweighting network trained on a small dataset or model can be fixed and reused for larger-scale training with negligible overhead.

**Strengths:**

- Instead of learning weights for per-sample parameters (which scales poorly with dataset size, as in SEAL ) or using large auxiliary models, ReBiSA learns a function that maps loss to importance, leading to its efficiency and transferability.
- The finding that the reweighting network is transferable across model scales (e.g., from TinyLlama-1.1B to Llama3-8B) and data scales (10k to 100k samples) demonstrates good effectiveness.

**Weaknesses:**

- The entire framework is optimized for safety by using a purely safe validation set. However, this raises a critical question: does this optimization for safety come at the cost of "helpfulness"? A common failure mode for safety-aligned models is "over-refusal" or "safety-taxing," where the model refuses to answer benign, non-harmful prompts. The paper provides no evaluation on general-purpose instruction-following or helpfulness benchmarks (e.g., MT-Bench, AlpacaEval). Without this, the full impact of ReBiSA on model utility is unknown.

- The design choices seem arbitrary and are not well justified. For instance, the reweighting network is a "two-layer feedforward network with a hidden layer of 100 units". Why 100 units? Why two layers? Would a simpler logistic regression (one layer) work? Is performance sensitive to this architecture?

**Questions:**

Please refer to the weaknesses.

---

### Official Review · Reviewer_Bhof · 2025-11-04

**Soundness:** 2
**Presentation:** 2
**Contribution:** 2
**Rating:** 4
**Confidence:** 4

**Summary:**

This paper addresses the safety alignment of LLMs. It takes a data-centric approach and designs an algorithm to filter out the unsafe data in the fine-tuning set. The proposed algorithm is built on some existing works on bilevel data selection [shen et al., sow et al.] by generalizing the data selection function to neural network parameterized ones. Empirical evidence suggests the nn parameterization improves the performance over non-parameterized baseline.

**Strengths:**

1. The paper addresses the data selection problem for LLM fine-tuning, which is a highly relevant and important topic.
2. The paper is overall clear and easy to follow.

**Weaknesses:**

1. The novelty might be limited. The proposed algorithm largely follows existing algorithms, with the only difference being the DNN parameterization of the scoring function. I would suggest looking into improvements over the existing bilevel data selection algorithms, .e.g, is there better formulation for the bilevel data selection problem? How can we design a bilevel data selection algorithm to balance the memory cost and solution quality?

2. The experimental results lack some critical baselines. In Table 1, It might be beneficial to report the average performance of a random selection baseline. There are some recent works reporting that random selection is able to beat most data selection algorithms [1]. Thus a thorough comparison can prove beneficial to this line of work.

[1] Tingyu Xia et al., Rethinking Data Selection at Scale: Random Selection is Almost All You Need.

**Questions:**

My main concerns are listed in the weaknesses section.

---

### Note · Authors · 2025-11-12

I have read and agree with the venue's withdrawal policy on behalf of myself and my co-authors.